# EnergyHair: Sketch-Based Interactive Guide Hair Design Using Physics-Inspired Energy

Yuanwei Zhang*
The University of Tokyo

Shinichi Kinuwaki†
Unaffiliated

Nobuyuki Umetani‡
The University of Tokyo

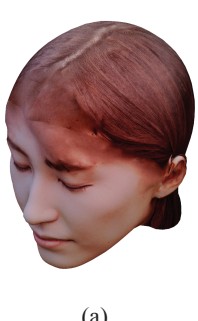 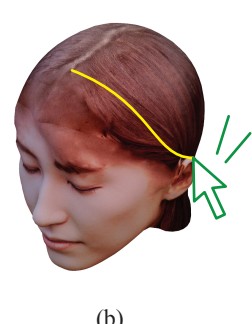 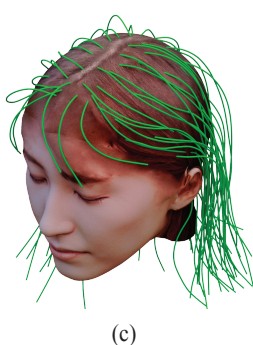 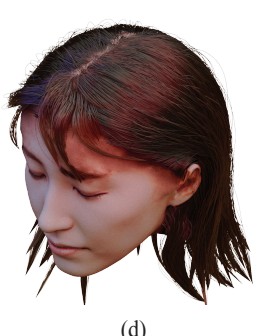

(a)         (b)         (c)         (d)

Figure 1: We present a sketching interface to design guide hair for character modeling using multi-view stereo. Starting from the input head model (a), the user specifies the shape by sketching (b). The three-dimensional shape of the hair is obtained from a two-dimensional sketching input via the optimization of a physically-based elastic potential energy. The resulting guide hair (c) is used for the generation of entire hair strands (d).

## ABSTRACT

This study introduces a new sketch-based hairstyle authoring interface for virtual characters captured using multi-view stereo. Individual hairs are typically modeled by interpolating guide hairs, which are downsampled hair strands representing the overall hair shapes. Unlike conventional geometric modeling, we incorporate the physical properties of hairs, such as gravity, collision, and bending resistance, in guide hair modeling. The use of physics-related shape optimization in the interface allows natural-looking 3D hair shapes to be modeled from minimal user specifications. We provide an interactive sketch tool that allows the user to specify hair shapes from multiple viewing angles. Our interface generates feedback regarding how much a designed sketch conforms to the laws of physics, enabling the user to strike a balance between artistic intention and physical naturalness. We further introduce an automatic sketch suggestion system based on the hair orientation obtained using image filtering.

**Index Terms:** Computing methodologies—Computer graphics—Shape modeling; Computing methodologies—Computer graphics—Graphics systems and interfaces; Computing methodologies—Computer graphics—Animation—Physical simulation

## 1 INTRODUCTION

The capture of real human actors as virtual avatars has become popular in the entertainment industry owing to the development of multi-view reconstruction techniques. Although current technology allows faithful reconstruction of the face to the extent that it is indistinguishable from the real actor, the accurate reproduction of hair remains a challenge, as hair is highly specular and typically thinner than the image resolution. Furthermore, owing to occlusion,

*e-mail: yuanwei.zh@gmail.com
†e-mail: shinichikinuwaki@gmail.com
‡e-mail: n.umetani@gmail.com

it is difficult to estimate the shape of entire 3D hair strands based on outside appearance. Therefore, hair is typically designed manually by artists on a bald-headed model with captured facial geometry.

The approximately 100 representative hair strands used for interpolation of other hair shapes are known as guide hairs. Several commercial hair editing tools, such as Ornatrix Maya [10] and Maya XGen [1], provide an interface for designing hairstyles by *geometric* editing guide hairs. However, during the editing process, hair often takes physically unreasonable shapes, such as those penetrating the head mesh or defying gravity. This is because the basic physics of hair (e.g., gravity, bending stiffness, and collision) are ignored in geometric editing. Furthermore, the user typically has to perform tedious manual tasks and frequently change the viewpoint to ensure that the 3D hairs look natural from multiple viewing angles.

Although most high-end 3D capturing systems faithfully replicate complicated hairstyles [2, 12, 16, 22, 24, 28, 41], it is difficult for designers to interactively control the design of the overall hairstyle using densely captured hair. The learning-based approach [5, 17, 31, 33, 43] enables hair to be synthesized from limited user input; however, the hair shape is restricted to what is presented in the dataset. We draw insights from *Skippy* [20], which allows users to draw 3D curves around a given geometry from a single-view 2D sketching input, and extend the idea to a multi-view sketch input.

Our key contribution is the implementation of a surrogate physics simulation, which may be inaccurate, but captures the fundamental property of deformation inside the sketch-based modeling system. Our system allows the user to sketch hair on top of the captured head shape. During sketching, the system concurrently optimizes the 3D shape of the hair and the physical parameters in the background to ensure that the hair fits into the sketch while satisfying the laws of physics as accurately as possible from the limited user shape specifications. The degree of physical inaccuracy in the hair design is shown as feedback to the user, and the program allows the user to make a trade-off between design intentions and physical plausibility. We further propose an algorithm to automatically suggest a user's drawing while the user is sketching.

## 2 RELATED WORK

Hair modeling has been a subject of study in the computer graphics field for many years. One such study was conducted by Ward et al. [35], who surveyed hair modeling and simulation techniques. More recently, state-of-the-art technologies have further pushed the limits of hair modeling.

**Hair Capture**   Paris et al. [27] introduced orientation fields as a way to capture hair strands, where controlled lighting was used to determine the strands' orientation. Inspired by the orientation fields used in [27], many other studies have similarly extended hair capturing technology. For instance, a handheld or video camera was used under natural lighting conditions in [36] to further challenge the elaborated lighting setup in [27]. Paris et al. [28] later extended their previous work [27] by introducing the use of structure tensors to infer the hidden geometry between the hair surface and scalp, which enabled the modeling of hair inside the surface. Luo et al. [21] considered highlighted hair strands to compute the orientation fields. Subsequently, a set of partial depth maps of hair was constructed by enforcing the structural continuities of the orientation field. The line-based PatchMatch Multi-view stereo (MVS) method presented in [24] reconstructs 3D line segments instead of 3D plane patches, which addresses the deficiencies of the conventional MVS method for modeling hair. However, these methods only consider the geometry of hair and ignore the underlying physics.

Hu et al. [13] captured a simulation-ready hair model by computing three fields: orientation, silhouette, and the motion of every frame. Then, an energy function built upon these three fields was optimized to compute physics-based parameters so that the generated hair model could be used in the simulation. However, the method described in this paper does not allow artists to change the captured hair model, as the computation is too expensive for an interactive system.

**Hair Simulation**   Hair simulation has been studied for several years in the computer graphics community. An early approach proposed by Rosenblum et al. [29] applies mass-spring models to individual hairs. The approach uses a linear spring for the stretch and an angular spring between segments for the bend. A subsequent altitude mass-spring model described in [32] simulates all individual hair strands with bending and twisting energy instead of considering hair as clumped strands. The elastic rod model can also be used for hair simulation. Pai [26] introduced a modeling primitive called a strand, which is based on Cosserat rod theory. Extending from the strand model, the super-helix model in [4] models curly hairs with a piecewise helical structure. Although these studies allow CG artists to simulate hair animation, they are not intended for use in interactive modeling.

**Sketch-based Hair Design**   A survey on sketch-based modeling [25] notes that hair is notoriously difficult to model owing to the sheer number of elements. Xing et al. [38] introduced a 3D sketching system using virtual reality technology to model hair geometry. However, because access to VR devices is currently limited, modeling on a computer screen remains the mainstream approach. To accommodate this, direct reconstruction of hair models from 2D sketches has been studied. Chen et al. [8] described a generative sketch model for human hair analysis and synthesis by treating hair images as 2D piecewise smooth vectors. Malik et al. [23] presented interaction techniques and algorithms to quickly model detailed hairstyles using intuitive sketching interfaces. However, these studies are geometry-based, and do not leverage the underlying physics. A data-driven approach was introduced in [14, 15], in which user strokes serve as a guide to search for matching examples from the database.

Wither et al. [37] introduced a hair sketching system that estimates physical parameters such as hair strand length, curvature, and ellipticity from the shape of 2D sketches. Although this approach takes physics into consideration, the limited parameter space makes it difficult to precisely control the shape of individual hair strands.

## 3 USER INTERFACE

**Input**   The user starts by providing the set of images used for the multi-view stereo reconstruction $\{\mathbf{I}_1, \mathbf{I}_2, \ldots, \mathbf{I}_n\}$, the resulting reconstructed mesh of the head and the $4 \times 4$ homography transformation matrices that transform the 3D position of the head to the image coordinate for each image $\{\mathbf{H}_1, \mathbf{H}_2, \ldots, \mathbf{H}_n\}$. The head mesh of a model can be generated by deforming the template base mesh of the head such that it fits into the point set obtained by multi-view stereo. The fitted mesh has an accurate geometry and texture where the skin is exposed (i.e., the face), and the area covered by the hair, although inaccurate, roughly follows the shape of the scalp. We used the commercial software Wrap3 [30] for the fitting. The user also provides flags on the elements of the head mesh on which hair can grow.

**Initialization**   Guide hairs are typically designed so that their roots are uniformly distributed on the scalp. Because it is tedious to manually specify the root of the guide hairs, we automatically sampled the root's position based on Poisson disk sampling on the mesh. The user can move the hair at any time by dragging the root with a mouse. Using the physics simulation, we grew the guide hairs 5 cm from the root.

**View Rotation**   The user can freely change the point of view around the head by dragging the mouse. When the mouse is released, the view is snapped to the closest input image projection, $\mathbf{H}_i$, and the corresponding background of the image, $\mathbf{I}_i$, is displayed. We measured the closeness of the view transformation by the polar coordinate distance of the camera's focal point position relative to the center of the head. Typically, head images are taken from surrounding viewpoints with a similar radius and the same side of the camera up (see Figure 2, middle up). The polar coordinates provide a natural transition between view transformations.

**Sketch Authoring**   The user selects one hair and draws a curve to specify how it should look from the current viewpoint. While sketching, the interface suggests a stroke by detecting the hair orientation of the image and integrating it with the current mouse position. Although the guide hair orientation typically does not accurately match the underlying orientation of the hair image, this functionality helps the sketching process, as the hair image is typically occluded by the mouse cursor. The selected hair changes its shape and length to fit into the sketch. The user can sketch from multiple view angles, and the system attempts to fit the hair into sketches as closely as possible.

However, the sketch may be physically incorrect. For example, hair may penetrate the head mesh, exhibit cusps, or move upwards against gravity. Therefore, the program denotes physical plausibility by color for the selected hair. If physical plausibility is low, the user may employ the *relaxation* tool, which increases physical plausibility by automatically moving the sketch. The user may also drag the stroke manually to edit its shape. See Figure 3 for an illustration.

## 4 METHOD

Let $\mathcal{V}$ denote the set of parameters to represent the deformation of all hairs, $\Theta$ denote the parameters of the hair simulation, and $\mathcal{U}$ denote the parameters of the user's specification by sketch. Our physics-inspired design defines two energies: the *elastic potential energy* of the entire hair $W : \mathcal{V} \times \Theta \to \mathbb{R}$, and *sketching energy*, which penalizes the difference between the input sketch and hair shape projected on the screen $K : \mathcal{V} \times \mathcal{U} \to \mathbb{R}$.

We need to reconcile two conflicting paradigms: physical plausibility and artistic intention. To provide the user with an interface to interactively explore the trade-off between physics simulation and geometric shape editing, we propose to define the hair shape $\mathcal{V}$ as

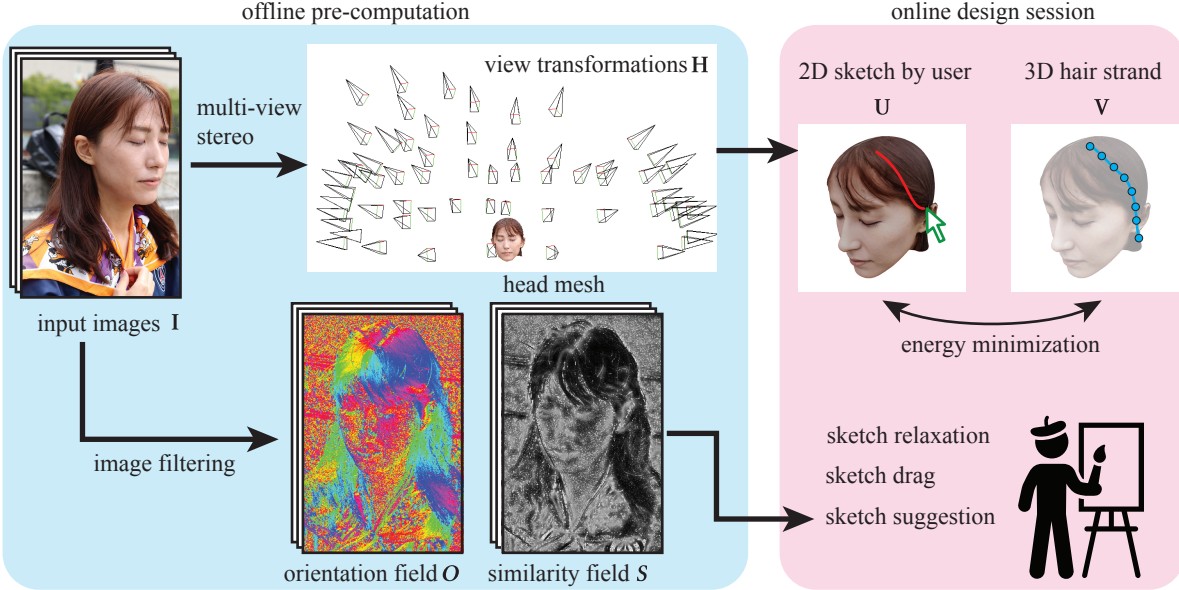

offline pre-computation

Figure 2: Workflow of our interface

the minimizer of the energy combining physics-related energy and sketch-related energy:

$$\arg\min_{\mathcal{V},\Theta} W(\mathcal{V},\Theta) + K(\mathcal{V},\mathcal{U}). \qquad (1)$$

When the hair shape $\mathcal{V}$ and physics parameter $\Theta$ are given by the energy minimization, the user interactively edits the sketch $\mathcal{U}$ while receiving feedback on its physical validity. The minimizer of (1) satisfies the following relation

$$\frac{\partial W}{\partial \mathcal{V}} = -\frac{\partial K}{\partial \mathcal{V}}, \qquad (2)$$

where the right side represents the artificial force resulting from the constraints specified by the user. Thus, we evaluated physical validity by measuring the magnitude of the right-hand side. Furthermore, in the relaxation operation, we optimized the sketch $\mathcal{U}$ such that the magnitude on the right-hand side is reduced.

### 4.1 Fitting Hair Into the Sketch

This section describes our definition of sketching energy $K$, and our method for providing an interface for editing the sketch. The total sketching energy $K$ is the sum of the sketching energies for each hair, $K'$. We denote hair as the sequence of 3D points $\mathbf{V} = (\mathbf{v}_1, \mathbf{v}_2, \ldots, \mathbf{v}_{|V|})$, where $\mathbf{v}_i \in \mathbb{R}^3$ and $|V|$ is the number of points in the hair. Assume that hair $\mathbf{V}$ has multiple sketches $\mathcal{U}_{\mathbf{V}} = (\mathbf{U}^1, \mathbf{U}^2, \ldots)$, where each sketch is drawn in the corresponding view transformation $(\mathbf{H}^1, \mathbf{H}^2, \ldots)$.

The sequence of 2D input points is provided by a user with a single stroke. Because this sequence contains various distances, we resample it with equal distance $L_{sketch}$, where we set the value to 5% of the screen width. Each sketch $\mathbf{U} \in \mathcal{U}_{\mathbf{V}}$ is a 2D point sequence $\mathbf{U} = (\mathbf{u}_1, \mathbf{u}_2, \ldots, \mathbf{u}_{|U|})$, where $\mathbf{u}_j \in \mathbb{R}^2$ and $|U|$ is the number of points. A hair deforms such that it fits into the user's sketch in the given multiple-view projections by adding the sketch energy to the physical energy

$$K'(\mathbf{V},\mathcal{U}_{\mathbf{V}}) = \sum_{\mathbf{U} \in \mathcal{U}_{\mathbf{V}}} \left\{ \mathcal{E}_{2D}(\mathbf{U}) + \sum_{\mathbf{v} \in \mathbf{V}} Dist^2(\mathbf{v},\mathbf{U}) \right\}, \qquad (3)$$

where $\mathcal{E}_{2D}$ denotes the elastic potential energy of a 2D elastic rod with a straight-rest configuration. We used this elastic energy as

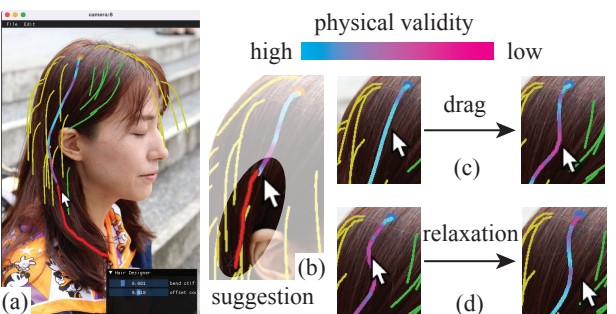

Figure 3: Interface of our tool: (a) screenshot of our tool; (b) suggestion of stroke during sketch; (c) dragging sketch; (d) sketch relaxation tool.

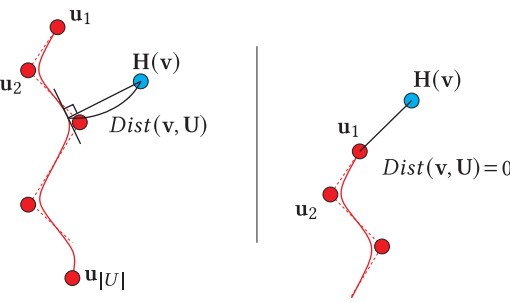

Figure 4: Distance between a vertex position of a hair on the screen $\mathbf{H}(\mathbf{v})$ and the sketch. The distance is computed by the nearest position on the quadratic B-spline curve, with control points given by the sketch $\mathbf{u}_i$ (left). When the nearest point is at the endpoints of the spline, we exclude the distance by setting it to zero (right).

the regularizer for the stroke control points. The elastic potential energy consists of the stretching and bending energies, which are computed on the polyline connecting points in $\mathbf{U}$. Stretching energy is defined as the sum of the squared edge length differences from $L_{sketch}$, whereas bending energy is defined as the squared sum of the discrete curvature of the polyline. The curvature was computed using a formulation of the *discrete elastic rod* [3], in which the discrete curvature is defined as $\kappa = 2\tan\theta/2$, where $\theta$ is the turning angle between adjacent edges.

$Dist(\mathbf{v}, \mathbf{U})$ is the distance between the sketch and hair computed in the 2D screen space. Because the polyline is not $C^1$ continuous, it is difficult to compute the closest point smoothly. Drawing inspiration from [19], which uses the spline curve for collision detection between yarns, and [40], which uses the loop subdivision surface on the triangle mesh to define proximity, we employed a spline representation to compute the closest point. Specifically, we used a quadratic B-spline curve with an open and uniform knot, with $\mathbf{U}$ as the control points.

Figure 4 illustrates the distance computation. Let $\alpha \in [0, 1]$ be the parameter of the spline curve where $\alpha = 0$ corresponds to the starting point of the sketch, and $\alpha = 1$ to the ending point. With this parameter, the 2D point on the spline is expressed as $Spline(\alpha, \mathbf{U})$. For each 3D point of hair $\mathbf{v}$, we find the parameter that results in the smallest distance.

$$\alpha_{min} = \arg\min_{0 \le \alpha \le 1} ||\mathbf{H}(\mathbf{v}) - Spline(\alpha, \mathbf{U})|| \qquad (4)$$

If the smallest distance was reached at the end of the sketch, we excluded the distance from the energy because the hair vertex was not covered by the sketch.

$$Dist(\mathbf{v}, \mathbf{U}) = \begin{cases} 0, & \text{if } \alpha_{min} \text{ is 0 or 1} \\ ||\mathbf{H}(\mathbf{v}) - Spline(\alpha_{min}, \mathbf{U})||, & \text{if } 0 < \alpha_{min} < 1 \end{cases} \qquad (5)$$

Note that this distance is not continuous, and parameter $\alpha$ is at the ends of the spline. However, this discontinuity rarely causes instability in energy minimization (1) using Newton's method.

The relaxation tool is implemented by reducing the energy in (3) in terms of the points of the sketch $\mathbf{U}$. Finding the exact minimum energy is not reasonable, as the user wants to retain their sketch to some extent. Therefore, we damped the energy minimization

$$\mathbf{U} = \arg\min_{\mathbf{U}} \left\{ K'(\mathbf{V}, \mathcal{U}_{\mathbf{V}}) + w_{damp}||\mathbf{U} - \bar{\mathbf{U}}||^2 \right\}, \qquad (6)$$

where $\mathbf{U}$ is the updated set of points in the sketch in the current view transformation, and $\bar{\mathbf{U}}$ is the coordinates before the update. We used the damping parameter $w_{damp} = 100$ in our study. When the sketch was dragged, we also minimized the energy in (6) using a hard constraint on the sketch's dragged vertex.

Hair Length Adjustment  It was difficult to adjust hair length during the simulation. The hair shape is influenced by the optimization result in (1), and the simulation is influenced by the length of the hair. Instability was observed when the length was automatically adjusted and the simulation did not converge. Therefore, we changed the hair length by adding or removing individual vertices until the simulation converged.

The length of hair was automatically adjusted such that it covered the sketch strokes of multiple views (see Figure 5). We projected the position of the hair tip onto the screen. The point on the screen was then projected onto the spline curve of the user's stroke. If the hair touched the stroke, the stroke parameter was determined. If the tip touched the stroke in the middle, we extended the hair. If the hair touched the stroke while the hair tip did not, we shortened the hair.

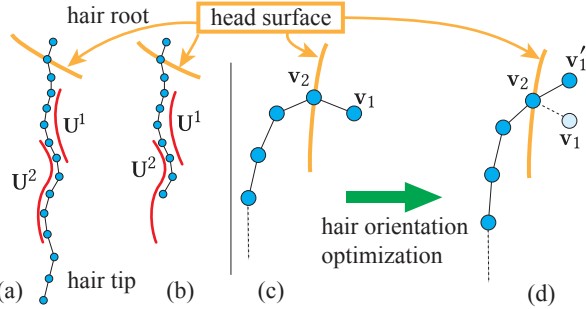

Figure 5: We shortened the hair if the tip of the hair was not present on the sketch (a) and elongated it (b) if the tip was in the middle of the sketch. Hair orientation is optimized by moving the first hair vertex $\mathbf{v}_1$ such that the first three hair vertices are aligned.

## 4.2  Automatic Sketch Suggestion

During the user's sketch, our interface suggests a sketch stroke shape based on the underlying image, $\mathbf{I}$. We prepared two fields: a 2D orientation field and a similarity field computed from each input image. The orientation field represents the orientation of the hair used to extend the stroke from the current mouse position. The similarity field, which represents the similarity between each part of the image and its surroundings, is used as a stopping criterion for stroke extension.

Orientation Field  We used an edge detection filter, specifically the Canny filter (i.e., the Gaussian first derivative), to compute our 2D orientation field following [27]. For example, the kernel of the Canny filter for horizontal edge detection can be written as $-x\exp\{-(x^2 + y^2)/\sigma^2\}$, where $x$ and $y$ are the horizontal and vertical image coordinate differences, respectively. Using the scale $\sigma^2 = 2$, we computed the filter for 18 directions uniformly sampled from 0 to $\pi$. Following Paris et al. [27], we determined the orientation that maximized the Canny filter output $\phi$ for each pixel. Although there are many other methods to compute orientation in a hair image, such as the DoG filter used in [21], Gabor kernel in [7, 18], and log-Gabor kernel in [34], we found that our choice is sufficient for our interface, since it is already the simplest way.

The orientation field directly generated from the Canny filter exhibits significant noise, which manifests as distracting noise in the proposed sketch. Therefore, we blurred the orientation field by applying a bilateral filter based on the field's variance $\mathbf{Z}$, as well as a Gaussian filter based on the distance in the image, to average the orientation field of the pixels. First, we represented the orientation using a complex number $c = \exp(2i\phi)$. This representation is convenient for blurring because it does not exhibit a discontinuity around $\phi = \pi$. Next, we computed the variance of the unblurred orientation field $\mathbf{Z}$ following the same method as in [27]. The blurring function applied to pixel coordinates $\mathbf{p}$ can be written as

$$O_{\mathbf{p}} = \sum_{\mathbf{q} \in N(\mathbf{p})} w_g(\mathbf{p}, \mathbf{q}) w_z(\mathbf{p}, \mathbf{q}) O_{\mathbf{q}}, \qquad (7)$$

where $w_g = \exp(-||\mathbf{p} - \mathbf{q}||^2/\sigma_g^2)$ is the weight of the Gaussian blur, $w_z = \exp(-(\mathbf{Z}_{\mathbf{q}}/\mathbf{Z}_{\mathbf{p}})^2/\sigma_z^2)$ is the weight of the bilateral filter, and $N(\mathbf{p})$ is the neighbor of the pixel $\mathbf{p}$. In this study, we set the filter sizes $\sigma_g$ and $\sigma_z$ to 2. Because the aim of our 2D hair strand growing algorithm is to assist the user in automatically drawing a rough 2D hair sketch, a high resolution is not necessary. Therefore, we downsampled the blurred orientation field by $16 \times 16$.

Similarity Field  We propose the use of a similarity field, which shows the consistency of the orientation field, to extract the hair

region. Similarity is defined as the difference in orientation between a pixel $p$ and its neighbor $q$ as follows:

$$S_p = \sum_{q \in N(p)} w_z(p,q)|O_p - O_q|, \qquad (8)$$

where $w_z$ is the same as in (7), and $|O_p - O_q|$ denotes the absolute value of the complex number of $O_p - O_q$. A higher $S_p$ indicates a lower similarity $p$. We observed that the similarity field indicates the hair region in an image because the hair region is highly consistent with the orientation field. Semi-automatic hair region segmentation [6, 22] and learning-based hair segmentation [5, 31, 39, 41, 42] also identify the hair region; however, these methods are complicated, whereas manual marking [9] is tedious. Therefore, our similarity field is suitable for satisfying our demand. Figure 6 shows an example of the orientation and similarity fields for a specific hair image.

**Sketch Suggestion** With orientation field $O$ and similarity field $S$, the system suggests a sketch by extending the existing sketch from the current mouse position $p_0$. The inset shows the algorithm process. The orientation fields provide two opposite directions for a given point. We chose the direction with a positive inner product with the current tangent vector of the sketch, and extended the sketch with length $L_{sketch}$ in that direction. We iterated this growing process until the similarity value was below the threshold of 0.8, which indicates that the blurred orientation is different from its neighbor by approximately 20 degrees.

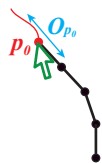

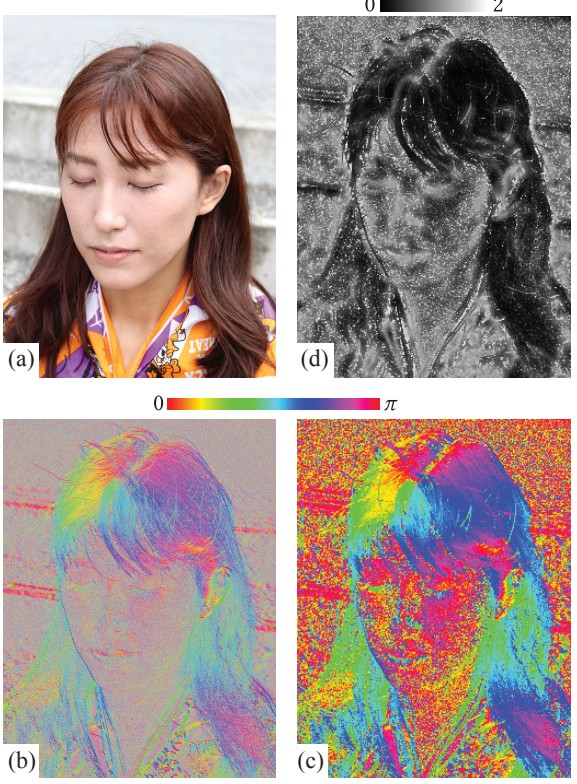

0 ▭ 2

0 ▭ $\pi$

Figure 6: An example of an orientation field and similarity field: (a) Original hair image. (b) Orientation field obtained from Canny filters. (c) Orientation field after blurring and downsampling. (d) Similarity field describing the differences between a pixel and its neighbors.

## 4.3 Hair Simulation

Given the physics parameter $\Theta$, the shape of hair strands $\mathcal{V}$ is simulated by minimizing the energy in (1). We defined the physics-related energy $W$ as the sum of the energies $W'$ defined independently for each hair strand $V$ as

$$W'(V) = \mathcal{E}_{3D}(V) + \sum_{v \in V} \left\{ g \cdot v + Depth^2(v) \right\}, \qquad (9)$$

where $g$ is the gravitational acceleration and $Depth : \mathbb{R}^3 \to \mathbb{R}^+$ is the penetration depth of a hair vertex into the head mesh. We use the adaptive distance field [11] to ensure fast computation. $\mathcal{E}_{3D}$ represents the elastic potential energy of the 3D rod. Similar to the sketching energy, we also considered the energy based on a discrete elastic rod [3] with a straight rest shape. We placed the first hair vertex of the point sequence $v_1$ inside the head mesh, and the second hair vertex $v_2$ on the surface. By changing the position of the first $v_1$, we can control the orientation of the hair growing from the head surface. Thus, we fixed the two vertices of the hair at the root (see Figure 5-right).

The Hessian matrix of (9) is a block pentadiagonal, where each block has a size of 3×3. The linear system for minimization using Newton's method can be trivially solved by the backward-forward weep algorithm. Note that we ignore inter-hair physics such as collision and adhesion. Note also that our goal is not to accurately simulate hair that faithfully reproduces the actual hair geometry, but rather to perform optimization in terms of hair shape and physics parameters at an interactive rate for interactive design.

The hair shape $\mathcal{V}$ and physics-related parameters $\Theta$ are *separately* optimized in (1). In other words, in each frame, we optimize for the hair shape by solving (2), and then optimize the physics-related parameters while the hair shape $\mathcal{V}$ remains fixed. In this study, we optimized the gravitational acceleration $g$ and the growth orientation for each hair. The direction of the gravitational acceleration needs optimization because it is typically unknown in the model captured using MVS. To optimize the gravitational acceleration coefficient $g$, we moved it toward the averaged direction of the hairs, where direction of each individual hair was computed as the position of the center of gravity relative to the root. The orientation of hair growth is updated for each hair $V$ by moving the first vertex $v_1$ toward the position, thus aligning the first three vertices (see Figure 5-right). To prevent hair from sagging completely under gravity, we limited the orientation of hair growth below the $\pi/2$ difference from the normal orientation of the head mesh.

## 5 IMPLEMENTATION AND RESULT

Our guide hair modelling tool was implemented as a single-threaded standalone C++ application. We used OpenGL for 3D graphics and the ImGui library for the user interface. Our design tool was developed in-house, and includes a linear solver.

**Performance** Our design tool was tested on a 16-inch 2019 model MacBook Pro. Our tool runs at 58 frames per second for 40 strands with 450 hair vertices, and 46 frames per second for 100 strands with 1035 hair vertices. The code used to filter the image for the computation of orientation and similarity fields was written in Python using the PyTorch library. For an image with a resolution of $6000 \times 4000$, generating a 2D orientation field costs approximately 90 s, blurring costs approximately 200 s, and downsampling and computing the similarity field cost 5 s. Total preprocessing requires no more than 5 min for a single image. All numbers were obtained from the computations on the CPU.

Figure 7 compares the distance functions that measure the difference between sketch and the guide hair. As for the naïve baseline distance function, we use the polyline representation instead of our B-Spline representation in (5). Specifically, each vertex of hair is projected on the screen $H(v)$, then the shortest distance to the

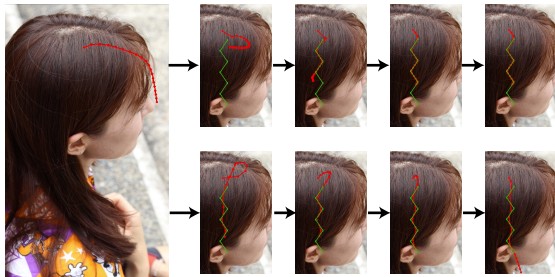

Figure 7: Comparison of the convergence of the guide hair deformation in different distance functions that compare the sketch (green line) and the hair (red line). The upper row shows the naïve distance function and the lower row shows our distance function.

polyline of sketch is computed. Given the zigzag sketch shape, the naïve approach failed to converge into the global minimum since its distance function is not smooth. The vertex is trapped at the crease of the polyline. On the other hands, our approach that approximate the input sketch with B-Spline curve successfully achieve global minimum where the entire sketch is covered with the hair on the screen.

Figure 1 and Figure 8 showcase the results created by the authors using our tool. After we designed the guide hair using our tool, we used Maya XGen to synthesize the other hair. During synthesis, we applied noise to ensure a natural look to the hair.

We informally evaluated our tool with the help of two professional animators who have experience using Maya XGen to create hair for digital characters. One participant mentioned that use of the existing tool requires a significant amount of tedious work to ensure that the 3D hair geometry looks physically reasonable without penetration between the head mesh and natural sagging under gravity. In addition, although this tool drastically accelerates the design of guide hair in the early prototyping stage, the fine-tuning of results is difficult in the final stage because the tool does not provide the final hair results generated from the guide hair. In future studies, we will generate a real-time preview of the final hair synthesis from guide hair.

## 6 LIMITATIONS AND FUTURE WORK

Although we demonstrated that our tool enables users to create complex 3D geometries of hair, there are a few limitations that remain to be addressed in future work.

Our system accounts for physics such as gravity, elasticity, and collisions. However, the physics model we applied was rather simple, as it assumes a straight-rest configuration of hair. Modeling curly hair is difficult because the underlying physics model suggests straight hair. We plan to use a more complicated hair physics model that encompasses hair with a helical rest configuration.

Currently, we only support the online optimization of gravity acceleration and hair orientation at the root. These physics-related parameters can reduce the physics-related energy with geometrical operations. The inclusion of additional physics-related parameters in online optimization, such as the bending stiffness of hair or curved rest shapes, is a topic for future studies.

Based on feedback from the artists, we are currently developing a real-time preview of the synthesized results of the hair, which will prove helpful for fine-tuning the guide hair design. We are aiming to move the synthesized hair together with the guide hair by defining frames on each vertex of the guide hair using parallel transport [3] and rigging the synthesized hair.

## ACKNOWLEDGEMENT

We would like to thank the anonymous reviewers for their suggestions and comments. This work was supported by JSPS KAKENHI Grant Number 21K11910.

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
