# OpenReview forum: "EnergyHair: Sketch-Based Interactive Guide Hair Design Using Physics-Inspired Energy"
_graphicsinterface.org/Graphics_Interface/2022/Conference — GI 2022_

### Official Review · Reviewer_BM5v · 2022-04-09
**Useful Tool for Guide Hair Design But**

**Rating:** 6
**Confidence:** 2

**Review:**

This paper proposes a sketch-based interactive system to help with guide hair design for commercial hair modeling tools. The baseline seems to be these commercial products where no guide hair suggestions or physics are considered at all. However, I feel a stronger baseline could be compared with, such as Malik et al. or Wither et al. If direct comparison is not possible, maybe some components of various algorithms could be compared.

The overall idea is sound, but the technical details are not totally clear to me. For example, it is mentioned that the gravity acceleration is optimized for each guide hair. Shouldn't the gravity acceleration be a fixed value on earth? Also the abstract and introduction sounded like you optimize for gravity, collision, and bending, but in the end you only optimized for gravity and root orientation, right? Overall the mathematical notations used in the paper feel weird at times.

The writing and figures could also be improved. A thorough editing pass is recommended.

---

### Official Review · Reviewer_gctL · 2022-04-13
**Transforming artist sketch of hair strains into drape, simulated hair on a 3d model using a physics-based method with artist-controlled regularizers**

**Rating:** 7
**Confidence:** 4

**Review:**

The authors present a tool to for artists generate hair strands on a character using their interactive sketch-based framework. The build a 3d reconstruction of a capture subject from multi-view stereo images and use this as a guide geometry to drape hair strands from. The artist creates sketches for hair strands using 2d projections, which drives simulated hair strands to closely align to the sketch following an energy minimization technique. The 3D hair is simulated using a standard elastic rod method, combined with a gravity term and a collision response from the 3D reconstruction head. The sketch also drive the hair, using a ICP-like term to drive the vertices of the hair to a spline-based representation of the sketch curve. The system will provide 'suggestions' for the sketch path based on information gleaned from processing the multi-view images -- orientation and similarity fields to direct the sketch path.

Although the paper does rely on existing methods for the sub-components, the novelty of the paper is the combination of these existing methods to create the system to perform the hair authoring. The energy minimization terms are well thought out and the sketch suggestion idea is good. The paper is well written with the contributions clearly explained throughout the exposition sections.

The paper seems like a pretty clear accept.

---

### Official Review · Reviewer_1Ksb · 2022-04-14
**Overall a decent paper, with flaws in the experiments/results section mainly, more testing would have been better.**

**Rating:** 8
**Confidence:** 4

**Review:**



The performance section was not well described, and a macbook is an odd choice of development platform - what processor, what memory, any GPU additions, the size of the screen is not relevant. As it's single threaded, was all the CPU used (typically now with multicore it would only use around 20-25% of the CPU in this case). The results in general are not very extensive, and typically preliminary - a more extensive comparison using a more linear set of hair strands would have been better.

Use of professional animators is not very easy for comparison, and you have no discussion about the evaluation metrics used (or even if they were tested) - this appears to be more of a "think-aloud" test.

It was not clear where XGen was used/compared. This needs to be made clearer (are we looking art comparison images). In Figure 8, it's not clear what stage is what stage, this could also be more clearly presented.

Figure 6 would have been better if it was bigger,

Otherwise the work is described well, can be reproduced. Comparisons with other systems is good and within reason this method moves things forward. I would like to have seen more comparison with XGen though, even being able to bring in the model generated and use XGen in this way ... the authors may have done this, but it's not clear.

---

### Decision · Program_Chairs · 2022-04-17

Accept